# Learning Partial Equivariances from Data

**David W. Romero**[*]
Vrije Universiteit Amsterdam
Amsterdam, The Netherlands
d.w.romeroguzman@vu.nl

**Suhas Lohit**
Mitsubishi Electric Research Laboratories
Cambridge, MA, USA
slohit@merl.com

## Abstract

Group Convolutional Neural Networks (G-CNNs) constrain learned features to respect the symmetries in the selected group, and lead to better generalization when these symmetries appear in the data. If this is not the case, however, equivariance leads to overly constrained models and worse performance. Frequently, transformations occurring in data can be better represented by a subset of a group than by a group as a whole, e.g., rotations in $[-90°, 90°]$. In such cases, a model that respects equivariance *partially* is better suited to represent the data. In addition, relevant transformations may differ for low and high-level features. For instance, full rotation equivariance is useful to describe edge orientations in a face, but partial rotation equivariance is better suited to describe face poses relative to the camera. In other words, the optimal level of equivariance may differ per layer. In this work, we introduce *Partial G-CNNs*: G-CNNs able to learn layer-wise levels of partial and full equivariance to discrete, continuous groups and combinations thereof as part of training. Partial G-CNNs retain full equivariance when beneficial, e.g., for rotated MNIST, but adjust it whenever it becomes harmful, e.g., for classification of 6 / 9 digits or natural images. We empirically show that partial G-CNNs pair G-CNNs when full equivariance is advantageous, and outperform them otherwise.[2]

## 1 Introduction

The translation equivariance of Convolutional Neural Networks (CNNs) [31] has proven an important inductive bias for good generalization on vision tasks. This is achieved by restricting learned features to respect the translation symmetry encountered in visual data, such that if an input is translated, its features are also translated, but not modified. Group equivariant CNNs (G-CNNs) [7] extend equivariance to other symmetry groups. Analogously, they restrict the learned features to respect the symmetries in the group considered such that if an input is transformed by an element in the group, e.g., a rotation, its features are also transformed, e.g., rotated, but not modified.

Nevertheless, the group to which G-CNNs are equivariant must be fixed prior to training, and imposing equivariance to symmetries not present in the data leads to overly constrained models and worse performance [6]. The latter comes from a difference in the data distribution, and the family of distributions the model can describe. Consequently, the group must be selected carefully, and it should correspond to the transformations that appear naturally in the data.

Frequently, transformations appearing in data can be better represented by a subset of a group than by a group as a whole, e.g., rotations in $[-90°, 90°]$. For instance, natural images much more likely show an elephant standing straight or slightly rotated than an elephant upside-down. In some cases, group transformations even change the desired model response, e.g., in the classification of the digits 6 and 9. In both examples, the data distribution is better represented by a model that respects rotation equivariance *partially*. That is, a model equivariant to some, but not all rotations.

36th Conference on Neural Information Processing Systems (NeurIPS 2022).

---

[*]Work done at Mitsubishi Electric Research Laboratories.

[2]Our code is publicly available at github.com/merlresearch/partial_gcnn.

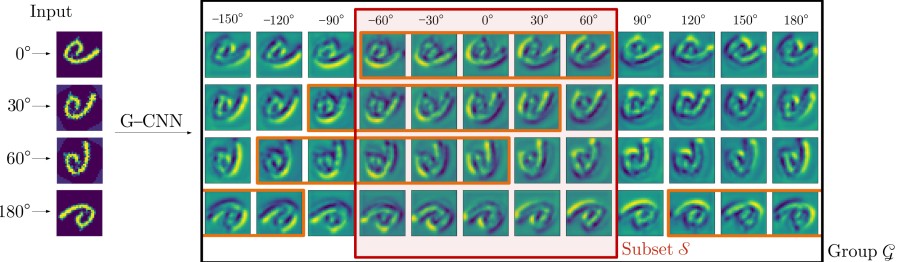

Figure 1: Partial group convolution. In a group convolution, the domain of the output is the group $\mathcal{G}$. Consequently, all output components are part of the output for any group transformation of the input. In a partial group convolution, however, the domain of the output is a learned subset $\mathcal{S}$ and all values outside of $\mathcal{S}$ are discarded. As a result, parts of the output in a partial group convolution can change for different group transformations of the input. In this figure, the output components within $\mathcal{S}$ for a $0°$ rotation (outlined in orange) gradually leave $\mathcal{S}$ for stronger transformations of the input. For strong transformations –here a $180°$ rotation–, the output components within $\mathcal{S}$ are *entirely* different. This difference allows partial group convolutions to distinguish among input transformations. By controlling the size of $\mathcal{S}$, the level of equivariance of the operation can be adjusted.

Moreover, the optimal level of equivariance may change per layer. This results from changes in the likelihood of some transformations for low and high-level features. For instance, whereas the orientations of edges in an human face are properly described with full rotation equivariance, the poses of human faces relative to the camera are better represented by rotations in a subset of the circle.

The previous observations indicate that constructing a model with different levels of equivariance at each layer may be advantageous. Weiler and Cesa [46] empirically observed that manually tuning the level of equivariance at different layers leads to accuracy improvements for non-fully equivariant tasks. Nevertheless, manually tuning layer-wise levels of equivariance is not straightforward and requires iterations over several possible combinations of equivariance levels. Consequently, it is desirable to construct a model able to learn optimal levels of equivariance directly from data.

In this work, we introduce *Partial Group equivariant CNNs* (Partial G-CNNs): a family of equivariant models able to *learn* layer-wise levels of equivariance directly from data. Instead of sampling group elements uniformly from the group during the group convolution –as in G-CNNs–, Partial G-CNNs learn a probability distribution over group elements at each group convolutional layer in the network, and sample group elements during group convolutions from the learned distributions. By tuning the learned distributions, Partial G-CNNs adjust their level of equivariance at each layer during training.

We evaluate Partial G-CNNs on illustrative toy tasks and vision benchmark datasets. We show that whenever full equivariance is beneficial, e.g., for rotated MNIST, Partial G-CNNs learn to remain fully equivariant. However, if equivariance becomes harmful, e.g., for classification of 6 / 9 digits and natural images, Partial G-CNNs learn to adjust equivariance to a subset of the group to improve accuracy. Partial G-CNNs improve upon conventional G-CNNs when equivariance reductions are advantageous, and match their performance whenever their design is optimal.

In summary, our **contributions** are:

- We present a novel design for the construction of equivariant neural networks, with which layer-wise levels of partial or full equivariance can be learned from data.
- We empirically show that Partial G-CNNs perform better than conventional G-CNNs for tasks for which full equivariance is harmful, and match their performance if full equivariance is beneficial.

## 2 Background

This work expects the reader to have a basic understanding of concepts from group theory such as groups, subgroups and group actions. Please refer to Appx. A if you are unfamiliar with these terms.

**Group equivariance.** Group equivariance is the property of a map to respect the transformations in a group. We say that a map is equivariant to a group if whenever the input is transformed by elements of the group, the output of the map is equally transformed but not modified. Formally, for a group $\mathcal{G}$ with elements $g \in \mathcal{G}$ acting on a set $\mathcal{X}$, and a map $\phi : \mathcal{X} \rightarrow \mathcal{X}$, we say that $\phi$ is equivariant to $\mathcal{G}$ if:

$$\phi(g \cdot x) = g \cdot \phi(x), \quad \forall x \in \mathcal{X}, \ \forall g \in \mathcal{G}. \tag{1}$$

For example, the convolution of a signal $f : \mathbb{R} \to \mathbb{R}$ and a kernel $\psi : \mathbb{R} \to \mathbb{R}$ is *translation equivariant* because $\mathcal{L}_t(\psi * f) = \psi * \mathcal{L}_t f$, where $\mathcal{L}_t$ translates the function by $t$: $\mathcal{L}_t f(x) = f(x - t)$. That is, if the input is translated, its convolutional descriptors are also translated but not modified.

**The group convolution.** To construct neural networks equivariant to a group $\mathcal{G}$, we require an operation that respects the symmetries in the group. The *group convolution* is such a mapping. It generalizes the convolution for equivariance to general symmetry groups. Formally, for any $u \in \mathcal{G}$, the group convolution of a signal $f : \mathcal{G} \to \mathbb{R}$ and a kernel $\psi : \mathcal{G} \to \mathbb{R}$ is given by:

$$h(u) = (\psi * f)(u) = \int_{\mathcal{G}} \psi(v^{-1}u) f(v) \, d\mu_{\mathcal{G}}(v), \tag{2}$$

where $\mu_{\mathcal{G}}(\cdot)$ is the (invariant) Haar measure of the group. The group convolution is $\mathcal{G}$-equivariant in the sense that for all $u, v, w \in \mathcal{G}$, it holds that:

$$(\psi * \mathcal{L}_w f)(u) = \mathcal{L}_w (\psi * f)(u), \text{ with } \mathcal{L}_w f(u) = f(w^{-1}u).$$

**The lifting convolution.** Regularly, the input of a neural network is not readily defined on the group of interest $\mathcal{G}$, but on a sub-domain thereof $\mathcal{X}$, i.e., $f : \mathcal{X} \to \mathbb{R}$. For instance, medical images are functions on $\mathbb{R}^2$ although equivariance to 2D-translations and planar rotations is desirable. In this case $\mathcal{X} = \mathbb{R}^2$, and the group of interest is $\mathcal{G} = \text{SE}(2)$. Consequently, we must first *lift* the input from $\mathcal{X}$ to $\mathcal{G}$ in order to use group convolutions. This is achieved via the *lifting convolution* defined as:

$$(\psi *_{\text{lift}} f)(u) = \int_{\mathcal{X}} \psi(v^{-1}u) f(v) \, d\mu_{\mathcal{G}}(v); \quad u \in \mathcal{G}, v \in \mathcal{X}. \tag{3}$$

**Practical implementation of the group convolution.** The group convolution requires integration over a continuous domain and, in general, cannot be computed in finite time. As a result, it is generally approximated. Two main strategies exist to approximate group convolutions with regular group representations: group discretization [7] and Monte Carlo approximation [16]. The former approximates the group convolution with a fixed group discretization. Unfortunately, the approximation becomes *only* equivariant to the transformations in the discretization and not to the intrinsic continuous group.

A Monte Carlo approximation, on the other hand, ensures equivariance –in expectation– to the continuous group. This is done by uniformly sampling transformations $\{v_j\}, \{u_i\}$ from the group during each forward pass, and using these transformations to approximate the group convolution as:

$$(\psi \,\hat{*}\, f)(u_i) = \sum_j \psi(v_j^{-1}u_i) f(v_j) \mu_{\mathcal{G}}(v_j). \tag{4}$$

Note that this Monte Carlo approximation requires the convolutional kernel $\psi$ to be defined on the *continuous group*. As the domain cannot be enumerated, independent weights cannot be used to parameterize the convolutional kernel. Instead, Finzi et al. [16] parameterize it with a small neural network, i.e., $\psi(x) = \texttt{MLP}(x)$. This allows them to map all elements $v_j^{-1}u_i$ to a defined kernel value.

## 3 Partial Group Equivariant Networks

### 3.1 (Approximate) partial group equivariance

Before defining the partial group convolution, we first formalize what we mean by partial group equivariance. We say that a map $\phi$ is *partially equivariant* to $\mathcal{G}$, if it is equivariant to transformations in a subset of the group $\mathcal{S} \subset \mathcal{G}$, but not necessarily to all transformations in the group $\mathcal{G}$. That is, if:

$$\phi(g \cdot x) = g \cdot \phi(x) \quad \forall x \in \mathcal{X}, \forall g \in \mathcal{S}. \tag{5}$$

Different from equivariance to a *subgroup* of $\mathcal{G}$ –a subset of the group that also fulfills the group axioms–, we do not restrict the subset $\mathcal{S}$ to be itself a group.

As explained in detail in Sec. 3.3, partial equivariance holds, in general, *only approximately*, and it is exact only if $\mathcal{S}$ is a subgroup of $\mathcal{G}$. This results from the set $\mathcal{S}$ not being necessarily closed under group actions. In other words, partial equivariance is a relaxation of group equivariance similar to *soft invariance* [44]: the property of a map to be approximately invariant. We opt for the word *partial* in the equivariance setting to emphasize that (approximate) partial group equivariance arises by restricting the domain of the signals in a group convolution to a subset, i.e., a part, of the group.

### 3.2 The partial group convolution

Let $\mathcal{S}^{(1)}, \mathcal{S}^{(2)}$ be subsets of a group $\mathcal{G}$ and $\text{p}(u)$ be a probability distribution on the group, which is non-zero only on $\mathcal{S}^{(2)}$. The partial group convolution from a function $f : \mathcal{S}^{(1)} \to \mathbb{R}$ to a function

$h : \mathcal{S}^{(2)} \to \mathbb{R}$ is given by:

$$h(u) = (\psi * f)(u) = \int_{\mathcal{S}^{(1)}} \mathrm{p}(u) \psi(v^{-1} u) f(v) \, \mathrm{d}\mu_{\mathcal{G}}(v); \ u \in \mathcal{S}^{(2)}, v \in \mathcal{S}^{(1)}. \tag{6}$$

In contrast to group convolutions whose inputs and outputs are always defined on the entire group, i.e., $f, h : \mathcal{G} \to \mathbb{R}$, the domain of the input and output of the partial group convolution can also be *subsets of the group*. By learning these subsets, the model can become (*i*) fully equivariant $(\mathcal{S}^{(1)}, \mathcal{S}^{(2)} = \mathcal{G})$, (*ii*) partially equivariant $(\mathcal{S}^{(1)}, \mathcal{S}^{(2)} \neq \mathcal{G})$, or (*iii*) forget some equivariances $(\mathcal{S}^{(2)}$ a subgroup of $\mathcal{G})$.

### 3.3 From group convolutions to partial group convolutions

In this section, we show how group convolutions can be extended to describe partial equivariances. Vital to our analysis is the equivariance proof of the group convolution [7, 9]. In addition, we must distinguish between the domains of the input and output of the group convolution, i.e., the domains of $f$ and $h$ in Eq. 2. This distinction is important because they may be different for partial group convolutions. From here on, we refer to these as the ***input domain*** and the ***output domain***.

**Proposition 3.1.** *Let $\mathcal{L}_w f(u) = f(w^{-1} u)$. The group convolution is $\mathcal{G}$-equivariant in the sense that:*

$$(\psi * \mathcal{L}_w f)(u) = \mathcal{L}_w(\psi * f)(u), \textit{ for all } u, v, w \in \mathcal{G}. \tag{7}$$

*Proof.* [9]
$$(\psi * \mathcal{L}_w f)(u) = \int_{\mathcal{G}} \psi(v^{-1} u) f(w^{-1} v) \, \mathrm{d}\mu_{\mathcal{G}}(v) = \int_{\mathcal{G}} \psi(\bar{v}^{-1} w^{-1} u) f(\bar{v}) \, \mathrm{d}\mu_{\mathcal{G}}(\bar{v})$$
$$= (\psi * f)(w^{-1} u) = \mathcal{L}_w(\psi * f)(u).$$

In the first line, the change of variables $\bar{v} := w^{-1} v$ is used. This is possible because the group convolution is a map from the group to itself, and thus if $w, v \in \mathcal{G}$, so does $w^{-1} v$. Moreover, as the Haar measure is an invariant measure on the group, we have that $\mu_{\mathcal{G}}(v) = \mu_{\mathcal{G}}(\bar{v})$, for all $v, \bar{v} \in \mathcal{G}$. $\qquad\square$

**Going from the group $\mathcal{G}$ to a subset $\mathcal{S}$.** Crucial to the proof of Proposition 3.1 is the fact the group convolution is an operation from functions on the group to functions on the group. As a result, $w^{-1} u$ is a member of the output domain for any $w \in \mathcal{G}$ applied to the input domain. Consequently, a group transformation applied to the input can be reflected by an equivalent transformation on the output.

Now, consider the case in which the output domain is not the group $\mathcal{G}$, but instead an arbitrary subset $\mathcal{S} \subset \mathcal{G}$, e.g., rotations in $[-\frac{\pi}{2}, \frac{\pi}{2}]$. Following the proof of Proposition 3.1 with $u \in \mathcal{S}$, and $v \in \mathcal{G}$, we observe that the operation is equivariant to transformations $w \in \mathcal{G}$ as long as $w^{-1} u$ is a member of $\mathcal{S}$. However, if $w^{-1} u$ does not belong to the output domain $\mathcal{S}$, the output of the operation cannot reflect an equivalent transformation to that of the input, and thus equivariance is not guaranteed (Fig. 1). By tuning the size of $\mathcal{S}$, partial group convolutions can adjust their level of equivariance.

Note that equivariance is *only* obtained if Eq. 7 holds for *all* elements in the output domain. That is, if $w^{-1} u$ is a member of $\mathcal{S}$, for all elements $u \in \mathcal{S}$. For partial group convolutions, this is, in general, not the case as the output domain $\mathcal{S}$ is not necessarily closed under group transformations. Nevertheless, we can precisely quantify how much the output response will change for any input transformation given an output domain $\mathcal{S}$. Intuitively, this difference is given by the difference in the parts of the output feature representation that go in and out of $\mathcal{S}$ by the action of input group transformations. The stronger the transformation and the smaller the size of $\mathcal{S}$, the larger the equivariance difference in the output is (Fig. 1). The formal treatment and derivation are provided in Appx. B.1.

**Going from a subset $\mathcal{S}^{(1)}$ to another subset $\mathcal{S}^{(2)}$.** Now, consider the case in which the domain of the input and the output are both subsets of the group, i.e., $v \in \mathcal{S}^{(1)}$ and $u \in \mathcal{S}^{(2)}$. Analogous to the previous case, equivariance to input transformations $w \in \mathcal{G}$ holds at positions $u \in \mathcal{S}^{(2)}$ for which $w^{-1} u$ are also members of $\mathcal{S}^{(2)}$. Nevertheless, the input domain is not longer restricted to be closed, and thus the input can also change for different group transformations.

To see this, consider a partial group convolution from an input subset $\mathcal{S}^{(1)}$ to the group $\mathcal{G}$ (Fig. 2). Even if the output domain is the group, differences in the output feature map can be seen. This results from differences observed in the input feature map $f$ for different group transformations of the input.

Similar to the previous case, we can precisely quantify how much the output response changes for an arbitrary subset $\mathcal{S}^{(1)}$ in the input domain. Intuitively, the difference is given by the change in the parts of the input feature representation that go in and out of $\mathcal{S}^{(2)}$ by the action of the input group transformation. The stronger the transformation and the smaller the size of $\mathcal{S}^{(1)}$, the larger the difference in the output is (Fig. 2). The formal treatment and derivation of this quantity is provided in Appx. B.2.

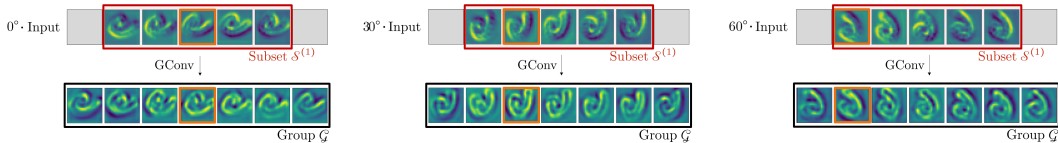

Figure 2: The effect of group subsets in the input domain. Partial group convolutions can receive input functions whose domain is not the group $\mathcal{G}$ but a subset $\mathcal{S}^{(1)}$, e.g., in a mid layer of a Partial G-CNN. Consequently, even if the output domain is the group, i.e., $\mathcal{S}^{(2)}=\mathcal{G}$, a partial group convolution does not produce exactly equivariant outputs by design. The difference from exact equivariance in the output response is proportional to the strength of the input transformation and the size of $\mathcal{S}^{(1)}$.

**Special case: lifting convolutions.** To conclude, we note that the lifting convolution (Eq. 3) is a special case of a partial group convolution with $\mathcal{S}^{(1)}=\mathcal{X}$ and $\mathcal{S}^{(2)}=\mathcal{G}$. Note however, that the lifting convolution is fully group equivariant. This is because the domain of the input $-\mathcal{X}-$ is closed under group transformations. Hence no input component leaves $\mathcal{X}$ for group transformations of the input.

### 3.4 Learning group subsets via probability distributions on group elements

So far, we have discussed the properties of the partial group convolution in terms of group subsets without specifying how these subsets can be learned. In this section, we describe how this can be done by learning a certain probability distribution on the group. We provide specific examples for discrete groups, continuous groups, and combinations thereof.

Vital to our approach is the Monte Carlo approximation to the group convolution presented in Sec. 2:

$$(\psi \hat{\star} f)(u_i) = \sum_j \psi(v_j^{-1} u_i) f(v_j) \bar{\mu}_{\mathcal{G}}(v_j).$$

As shown in Appx. C, this approximation is equivariant to $\mathcal{G}$ in expectation if the elements in the input and output domain are uniformly sampled from the Haar measure, i.e., $u_i, v_j \sim \mu_{\mathcal{G}}(\cdot)$.[3]

**Approach.** Our main observation is that we can prioritize sampling specific group elements during the group convolution by learning a probability distribution $\mathrm{p}(u)$ over the elements of the group. When group convolutions draw elements uniformly from the group, each group element is drawn with equal probability and thus, the resulting approximation is fully equivariant in expectation. However, we can also define a different probability distribution that draws some samples with larger probability. For instance, we can sample from a certain region, e.g., rotations in $\left[-\frac{\pi}{2}, \frac{\pi}{2}\right]$, by defining a probability distribution on the group $\mathrm{p}(u)$ which is uniform in this range, but zero otherwise. The same principle can be used to forget an equivariance by letting this distribution collapse to a single point, e.g., the identity, along the corresponding group dimension.

In other words, learning a probability distribution $\mathrm{p}(u)$ on the group that is non-zero *only* in a subset of the group can be used to effectively learn this subset. Specifically, we define a probability distribution $\mathrm{p}(u)$ on the output domain of the group convolution in order to learn a subset of the group $\mathcal{S}^{(2)}$ upon which partial equivariance is defined. Note that we only need to define a distribution on the output domain of each layer. This is because neural networks apply layers sequentially, and thus the distribution on the output domain of the previous layer defines the input domain of the next layer.

**Distributions for one-dimensional continuous groups.** We take inspiration from Augerino [3], an use the reparameterization trick [26] to parameterize continuous distributions. In particular, we use the reparameterization trick on the Lie algebra of the group [15] to define a distribution which is uniform over a connected set of group elements $[u^{-1}, \ldots, e, \ldots, u]$, but zero otherwise. To this end, we define a uniform distribution $\mathcal{U}(\mathfrak{u} \cdot [-1, 1])$ with learnable $\mathfrak{u}$ on the Lie algebra $\mathfrak{g}$, and map it to the group via the pushforward of the exponential map $\exp : \mathfrak{g} \to \mathcal{G}$. This give us a distribution which is uniform over a connected set of elements $[u^{-1}, \ldots, e, \ldots, u]$, but zero otherwise.[4]

For instance, we can learn a distribution on the rotation group $\mathrm{SO}(2)$, which is uniform between $[-\theta, \theta]$ and zero otherwise by defining a uniform probability distribution $\mathcal{U}(\theta \cdot [-1, 1])$ with learnable $\theta$ on the Lie algebra, and mapping it to the group. If we parameterize group elements as scalars $g \in [-\pi, \pi)$, the exponential map is the identity, and thus $\mathrm{p}(g)=\mathcal{U}(\theta \cdot [-1, 1))$. If we sample group

---

[3]Finzi et al. [16] show a similar result where $u_i$ and $v_j$ are the same points and thus $v_j \sim \mu_{\mathcal{G}}(\cdot)$ suffices.

[4]Note that an $\exp$-pushforwarded local uniform distribution is locally equivalent to the Haar measure, and thus we can still use the Haar measurement for integration on group subsets.

elements from this distribution during the calculation of the group convolution, the output domain will only contain elements in $[-\theta, \theta)$ and the output feature map will be partially equivariant.

**Distributions for one-dimensional discrete groups.** We can define a probability distribution on a discrete group as the probability of sampling from all possible element combinations. For instance, for the mirroring group $\{1, -1\}$, this distribution assigns a probability to each of the combinations $\{0,0\}, \{0,1\}, \{1,0\}, \{1,1\}$ indicating whether the corresponding element is sampled (1) or not (0). For a group with elements $\{e, g_1, \ldots, g_n\}$, however, this means sampling from $2^{n+1}$ elements, which is computationally expensive and potentially difficult to train. To cope with this, we instead define element-wise Bernoulli distributions over each of the elements $\{g_1, \ldots, g_n\}$, and learn the probability $p_i$ of sampling each element $g_i$. The probability distribution on the group can then be formulated as the joint probability of the element-wise Bernoulli distributions $\mathrm{p}(e, g_1, \ldots, g_n) = \prod_{i=1}^{n} \mathrm{p}(g_i)$.

To learn the element-wise Bernoulli distributions, we use the Gumbel-Softmax trick [25, 35], and use the Straight-Through Gumbel-Softmax to back-propagate through sampling. If all the probabilities are equal to 1, i.e., $\{p_i=1\}_{i=1}^{n}$, the group convolution will be fully equivariant. Whenever probabilities start declining, group equivariance becomes partial, and, in the limit, if all probabilities become zero, i.e., $\{p_i=0\}_{i=1}^{n}$, then only the identity is sampled and this equivariance is effectively forgotten.

**Probability distributions for multi-dimensional groups.** There exist several multi-dimensional groups with important applications, such as the orthogonal group $O(2)$ –parameterized by rotations and mirroring–, or the dilation-rotation group –parameterized by scaling and rotations–.

For multi-dimensional groups, we parameterize the probability distribution over the entire group as a combination of *independent probability distributions along each of the group axes*. For a group $\mathcal{G}$ with elements $g$ decomposable along $n$ dimensions $g=(d_1, ..., d_n)$, we decompose the probability distribution as: $\mathrm{p}(g) = \prod_{i=1}^{n} \mathrm{p}(d_i)$, where the probability $\mathrm{p}(d_i)$ is defined given the type of space – continuous or discrete–. For instance, for the orthogonal group $O(2)$ defined by rotations $r$ and mirroring $m$, i.e., $g = (r, m), r \in SO(2), m \in \{\pm 1\}$, we define the probability distribution on the group as $\mathrm{p}(g) = \mathrm{p}(r) \cdot \mathrm{p}(m)$, where $\mathrm{p}(r)$ is a continuous distribution, and $\mathrm{p}(m)$ is a discrete one as defined above.

### 3.5   Partial Group Equivariant Networks

To conclude this section, we illustrate the structure of Partial G-CNNs. We build upon Finzi et al. [16] and extend their continuous G-CNNs to discrete groups. This is achieved by parameterizing the convolutional kernels on (continuous) Lie groups, and applying the action of discrete groups directly on the group representation of the kernels.

In addition, we replace the isotropic lifting of Finzi et al. [16] with lifting convolutions (Eq. 3). Inspired by Romero et al. [39], we parameterize convolutional kernels as implicit neural representations with SIRENs [41]. This parameterization leads to higher expressivity, faster convergence, and better accuracy than ReLU, LeakyReLU and Swish `MLP` parameterizations used so far for continuous G-CNNs, e.g., [40, 16] –see Tab. 7, [27]-. The architecture of Partial G-CNNs is shown in Fig. 3.

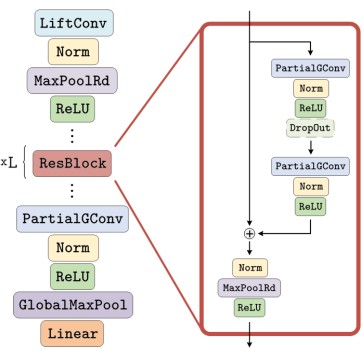

Figure 3: The Partial G-CNN.

## 4   Related work

**Group equivariant neural networks.** The seminal work of G-CNNs [7] has inspired several methods equivariant to many different groups. Existing methods show equivariance to planar [11, 50, 48], spherical rotations [47, 8, 12–14], scaling [49, 42, 38], and other symmetry groups [36, 4, 18]. Group equivariant self-attention has also been proposed [19, 37, 23]. Common to all these methods is that they are fully equivariant and the group must be fixed prior to training. Contrarily, Partial G-CNNs learn their level of equivariance from data and can represent full, partial and no equivariance.

**Invariance learning.** Learning the right amount of global invariance from data has been explored by learning a probability distribution over continuous test-time augmentations [3], or by using the marginal likelihood of a Gaussian process [44, 43]. Contrary to these approaches, Partial G-CNNs aim to learn the right level of equivariance at every layer and do not require additional loss terms. Partial G-CNNs relate to Augerino [3] in the form that probability distributions are defined on

Table 1: Results on MNIST6-180 and MNIST6-M.

| Base Group | Dataset | G-CNN | Partial G-CNN |
|---|---|---|---|
| SE(2) Mirroring | MNIST6-180 | 50.0 | **100.0** |
| | MNIST6-M | 50.0 | **100.0** |
| E(2) | MNIST6-180 | 50.0 | **100.0** |
| | MNIST6-M | 50.0 | **100.0** |

Table 2: Augerino vs. Partial G-CNNs. Terms in parentheses show accuracy of Partial G-CNNs.

| Base Group | No. Elems | Classification Accuracy (%) | | |
|---|---|---|---|---|
| | | RotMNIST | CIFAR10 | CIFAR100 |
| SE(2) | 8 | 99.17 (**99.23**) | 82.38 (**88.59**) | 52.98 (**57.26**) |
| | 16 | **99.25** (99.18) | 82.53 (**88.59**) | 51.47 (**57.31**) |
| E(2) | 8 | **99.12** (97.78) | 84.29 (**89.00**) | 52.59 (**55.22**) |
| | 16 | **99.18** (98.35) | 83.54 (**90.12**) | 54.76 (**61.46**) |

continuous groups. However, Partial G-CNNs are intended to learn partial layer-wise equivariances and are able to learn probability distributions on discrete groups. There also exist learnable data augmentation strategies [34, 20, 33, 5] that can find transformations of the input that optimize the task loss. We can also view Augerino as learning a smart data augmentation technique which we compare with. In contrast to these methods, Partial G-CNNs find optimal partial equivariances at each layer.

**Equivariance learning.** Learning equivariant mappings from data has been explored by meta-learning of weight-tying matrices encoding symmetry equivariances [51, 1] and by learning the Lie algebra generators of the group jointly with the parameters of the network [10]. These approaches utilize the same learned symmetries across layers. MSR [51] is only applicable to (small) discrete groups, and requires long training times. L-Conv [10] is only applicable to continuous groups and is not fully compatible with current deep learning components, e.g., pooling, normalization. Unlike these approaches, Partial G-CNNs can learn levels of equivariance at every layer, are fully compatible with current deep learning components, and are applicable for discrete groups, continuous groups and combinations thereof. We note, however, that Zhou et al. [51], Dehmamy et al. [10] learn the structure of the group from scratch. Contrarily, Partial G-CNNs start from a (very) large group and allows layers in the network to constrain their equivariance levels to better fit the data. Finzi et al. [17] incorporate soft equivariance constraints by combining outputs of equivariant and non-equivariant layers running in parallel, which incurs in large parameter and time costs. Differently, Partial G-CNNs aim to learn from data the optimal amount of partial equivariance directly on the group manifold.

# 5    Experiments

**Experimental details.** We parameterize all our convolutional kernels as 3-layer SIRENs [41] with 32 hidden units. All our networks –except for the (partial) group equivariant 13-layer CNNs [29] used in Sec. 5.1– are constructed with 2 residual blocks of 32 channels each, batch normalization [24] following the structure shown in Fig. 3. Here, we intentionally select our networks to be simple as to better assess the effect of partial equivariance. We avoid learning probability distributions on the translation part of the considered groups, and assume all spatial positions to be sampled in order to use fast `PyTorch` convolution primitives in our implementation. Additional experimental details such as specific hyperparameters used and complementary results can be found in Appx. E, F.

**Toy tasks: MNIST6-180 and MNIST6-M.** First, we validate whether Partial G-CNNs can learn partial equivariances. To this end, we construct two illustrative datasets: *MNIST6-180*, and *MNIST6-M*. *MNIST6-180* is constructed by extracting the digits of the class 6 from the MNIST dataset [31], and rotating them on the circle. The goal is to predict whether the number is a six, i.e., a rotation in $[-90°, 90°]$ was applied, or a nine otherwise. Similarly, we construct *MNIST6-M* by mirroring digits over the y axis. The the goal is to predict whether a digit was mirrored or not.

As shown in Tab. 1, G-CNNs are unable to solve these tasks as discrimination among group transformations is required. Specifically, SE(2)-CNNs are unable to solve MNIST6-180, and Mirror-CNNs –G-CNNs equivariant to reflections– are unable to solve MNIST6-M. Furthermore, E(2)-CNNs cannot solve any of the two tasks, because E(2)-CNNs incorporate equivariance to both rotations and reflections. Partial G-CNNs, on the other hand, easily solve both tasks with corresponding base groups. This indicates that Partial G-CNNs learn to adjust the equivariance levels in order to solve the tasks.

In addition, we verify the learned levels of equivariance for a Partial SE(2)-CNN on MNIST6-180. To this end, we plot the probability of assigning the label 6 to test samples of MNIST6-180 rotated on the whole circle. Fig 4 shows that the network learns to predict "6" for rotated samples in $[-90°, 90°]$, and "9" otherwise. Note that Partial G-CNNs learn the expected levels of partial equivariance without any additional regularization loss terms to encourage them –as required in Benton et al. [3]–.

Table 3: Accuracy on vision benchmark datasets.

| Base Group | No. Elems | Partial Equiv. | Classification Accuracy (%) | | |
|---|---|---|---|---|---|
| | | | RotMNIST | CIFAR10 | CIFAR100 |
| T(2) | 1 | - | 97.23 | 83.11 | 47.99 |
| SE(2) | 4 | ✗ | 99.10 | 83.73 | 52.35 |
| | | ✓ | **99.13** | **86.15** | **53.91** |
| | 8 | ✗ | 99.17 | 86.08 | 55.55 |
| | | ✓ | **99.23** | **88.59** | **57.26** |
| | 16 | ✗ | **99.24** | 86.59 | 51.55 |
| | | ✓ | 99.18 | **89.11** | **57.31** |
| E(2) | 8 | ✗ | **98.14** | 85.55 | 54.29 |
| | | ✓ | 97.78 | **89.00** | **55.22** |
| | 16 | ✗ | 98.35 | 88.95 | 57.78 |
| | | ✓ | **98.58** | **90.12** | **61.46** |

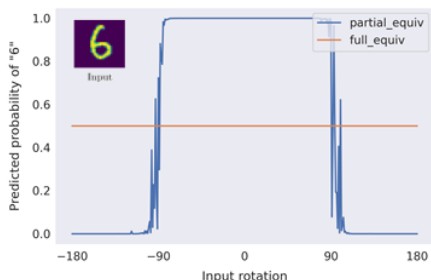

Figure 4: Learned equivariances for a "6" on MNIST6-180. Partial G-CNNs become equivariant to rotations on the semi-circle. Regular G-CNNs cannot solve the task.

**Benchmark image datasets.** Next, we validate Partial G-CNNs on classification datasets: RotMNIST [30], CIFAR-10 and CIFAR-100 [28]. Results on PatchCam [45] can be found in Appx. F.

We construct Partial G-CNNs with base groups $SE(2)$ and $E(2)$, and varying number of elements used in the Monte Carlo approximation of the group convolution and compare them to equivalent G-CNNs and ResNets (equivalent to $T(2)$-CNNs). Our results (Tab. 3) show that Partial G-CNNs are competitive with G-CNNs when full-equivariance is advantageous (rotated MNIST and Patch-Camelyon). However, for tasks in which the data does not naturally exhibit full rotation equivariance (CIFAR-10 and CIFAR-100), Partial G-CNNs consistently outperform fully equivariant G-CNNs.

**The need for learning partial layer-wise equivariances.** Next, we evaluate (*i*) the effect of learning partial equivariances instead of soft invariances, and (*ii*) the effect of learning layer-wise levels of equivariances instead of a single level of partial equivariance for the entire network.

For the former, we compare Partial G-CNNs to equivalent ResNets with Augerino [3] (Tab. 2). We extend our strategy to learn distributions on discrete groups to the Augerino framework to allow it to handle groups with discrete components, e.g., $E(2)$. For the latter, we construct regular G-CNNs and replace either the final group convolutional layer by a $T(2)$ convolutional layer, or the global max pooling layer at the end by a learnable MLP (Tab. 4). If determining the level of equivariance at the end of the network is sufficient, these models should perform comparably to Partial G-CNNs.

Tab. 2 shows that Augerino is competitive to Partial G-CNNs on rotated MNIST, but falls behind by a large margin on CIFAR-10 and CIFAR-100. This result can be explained by how these datasets are constructed. RotMNIST is constructed by rotating MNIST digits globally, thus it is not surprising that a model able to encode global invariances can match Partial G-CNNs. The invariance and equivariance relationships in natural images, however, are more complex, as they can be local as well. Consequently, tackling different levels of equivariance at each layer using Partial G-CNNs leads to benefits over using a single level of global invariance for the entire network.

Although the $T(2)$ and MLP alternatives outlined before could solve the MNIST-180 and MNIST-M toy datasets, we observed that Partial G-CNNs perform consistently better on the visual benchmarks considered (Tab. 4). This indicates that learning layer-wise partial equivariances is beneficial over modifying the level of equivariance only at the end of the model. In addition, it is important to highlight that Partial G-CNNs can become fully-, partial-, and non-equivariant during training. Alternative models, on the other hand, are either unable to become fully equivariant ($T(2)$ models) or very unlikely to do so in practice (MLP models).

**SIRENs as group convolution kernels.** Next, we validate SIRENs as parameterization for group convolutional kernels. Tab. 7 shows that $SE(2)$-CNNs with SIREN kernels outperform $SE(2)$-CNNs with ReLU, LeakyReLU and Swish kernels by a large margin on all datasets considered. This result suggests that SIRENs are indeed better suited to represent continuous group convolutional kernels.

## 5.1 Experiments with deeper networks

In addition to the simple networks of the previous experiments, we also explore partial equivariance in a group equivariant version of the 13-layer CNN of Laine and Aila [29]. Specifically, we construct partial group equivariant 13-layer CNNs using $SE(2)$ as base group, and vary the number of elements used in the Monte Carlo approximation of the group convolution. For each number of elements,

Table 4: Accuracy of G-CNNs with a MLP instead of global pooling, G-CNNs with a final $T(2)$-conv. layer, and our proposed Partial G-CNNs.

| BASE GROUP | NO. ELEMS | NET TYPE | CLASSIFICATION ACCURACY (%) | | |
|---|---|---|---|---|---|
| | | | ROTMNIST | CIFAR10 | CIFAR100 |
| SE(2) | 16 | T(2) | 99.04 | 82.76 | 52.51 |
| | | MLP | 99.00 | 86.25 | 56.29 |
| | | PARTIAL | **99.18** | **87.45** | **57.31** |
| E(2) | 16 | T(2) | 97.98 | 86.68 | 57.61 |
| | | MLP | **99.02** | 87.43 | 58.87 |
| | | PARTIAL | 98.58 | **90.12** | **61.46** |

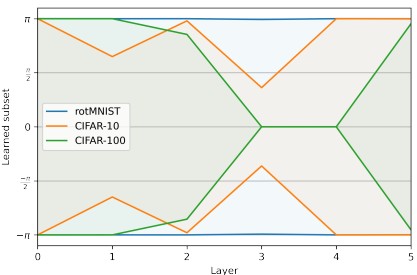

Figure 5: Example group subsets learned by Partial G-CNNs.

we compare Partial 13-layer G-CNNs to their fully equivariant counterparts as well as equivalent 13-layer CNNs trained with Augerino. Our results are summarized in Tab. 5.

On partial equivariant settings (CIFAR10 and CIFAR100), partial equivariant networks consistently outperform fully equivariant and Augerino networks for all number of elements used. Interestingly, translation equivariant CNNs outperform CNNs equivariant to $SE(2)$ on CIFAR10 and CIFAR100. This illustrates that overly restricting equivariance constraints can degrade accuracy. In addition, partial equivariant CNNs retain the accuracy of full equivariant networks in fully equivariant settings. By looking at the group subsets learned by partial equivariant networks (Fig 6), we corroborate that partial equivariant networks learn to preserve full equivariance if full equivariance is advantageous, and learn to disrupt it otherwise.

# 6 Discussion

**Memory consumption in partial equivariant settings.** G-CNNs fix the number of samples used to approximate the group convolution prior to training. In Partial G-CNNs we fix a maximum number of samples and adjust the number used at every layer based on the group subset learned. Consequently, a partial group convolution with a learned distribution $p(u){=}\mathcal{U}\big(\frac{\pi}{2}[-1,1]\big)$ uses half of the elements used in a corresponding group convolution. This reduction in memory and execution time leads to improvements in training and inference time for Partial G-CNNs on partial equivariant settings. We observe reductions up to 2.5× in execution time and memory usage on CIFAR-10 and CIFAR-100.

**Sampling per batch element.** In our experiments, we sample once from the learned distribution $p(u)$ at every layer, and use this sample for all elements in the batch. A better estimation of $p(u)$ can be obtained by drawing a sample per batch element. Though this method may lead to faster convergence and better estimations of the learned distributions, it comes at a prohibitive memory cost resulting from independent convolutional kernels that must be rendered for each batch element. Consequently, we use a single sample per batch at each layer in our experiments.

**Better kernels with implicit neural representations.** We replace $\mathrm{LeakyReLU}$, $\mathrm{ReLU}$ and $\mathrm{Swish}$ kernels used so far for continuous group convolution kernels with a SIREN [41]. Our results show that SIRENs are better at modelling group convolutional kernels than existing alternatives.

**Going from a small group subset to a larger one. What does it mean and why is it advantageous?** In Sec. 3.3 we described that a partial group convolution can go from a group subset $\mathcal{S}^{(1)}$ to a larger group subset $\mathcal{S}^{(2)}$, e.g., the whole group $\mathcal{G}$. Nevertheless, once a layer becomes partially equivariant, subsequent layers cannot become fully equivariant even for $\mathcal{S}^{(2)}{=}\mathcal{G}$. Interestingly, we observe that Partial G-CNNs often learn to disrupt equivariance halfway in the network, and return to the whole group afterwards (Fig. 5). As explained below, this behavior is actually advantageous.

Full equivariance restricts group convolutions to apply the same mapping on the entire group. As a result, once the input is transformed, the output remains equal up to the same group transformations. In partial equivariance settings, Partial G-CNNs can output different feature representations for different input transformations. Consequently, Partial G-CNNs can use the group dimension to encode different feature mappings. Specifically, some kernel values are used for some input transformations and other ones are used for other input transformations. This means that when Partial G-CNNs go back to a larger group subset from a smaller one, they are able to use the group axis to encode transformation-dependent features, which in turn results in increased model expressivity.

Table 5: Accuracy on vision benchmark datasets with (partial) group equivariant 13-layer CNNs [29].

| BASE GROUP | NO. ELEMS | PARTIAL EQUIV. | AUGERINO | CLASSIFICATION ACCURACY (%) | | |
|---|---|---|---|---|---|---|
| | | | | ROTMNIST | CIFAR10 | CIFAR100 |
| T(2) | 1 | - | - | 96.90 | 91.21 | 67.14 |
| SE(2) | 2 | ✗ | ✗ | 98.70 | 85.51 | 62.06 |
| | | | ✓ | **98.94** | 87.78 | 65.79 |
| | | ✓ | - | 98.72 | **92.48** | 66.72 |
| | 4 | ✗ | ✗ | 98.43 | 89.73 | 65.97 |
| | | | ✓ | **98.94** | 91.66 | 68.99 |
| | | ✓ | - | 98.78 | **92.28** | **69.83** |
| | 8 | ✗ | ✗ | 98.54 | 90.55 | 67.70 |
| | | | ✓ | **99.28** | 89.96 | 69.66 |
| | | ✓ | - | 98.77 | **91.99** | **70.80** |

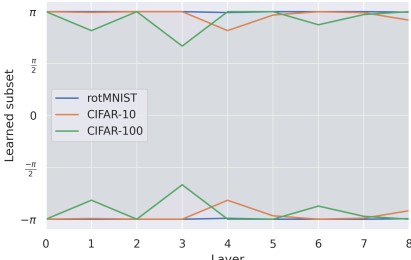

Figure 6: Group subsets learned by 13-layer Partial G-CNNs.

In Appx. F, we evaluate the effect of enforcing monotonically decreasing group subsets as a function of depth. That is, Partial G-CNNs whose subsets at deeper layers are equal or smaller than those at previous ones. Our results show that this monotonicity leads to slightly worse results compared to the unconstrained case, thus supporting the use of unconstrained learning of group subsets.

## 7 Limitations and future work

**Partial equivariances for other group representations.** The theory of learnable partial equivariances proposed here is only applicable to architectures using regular group representations, e.g., [7, 37]. Nevertheless, other type of representations exist with which exact equivariance to continuous groups can be obtained: irreducible representations [50, 48, 46]. We consider extending the learning of partial equivariances to irreducible representations a valuable extension of our work.

**Unstable training on discrete groups.** Although we can model partial equivariance on discrete groups with our proposed discrete probability distribution parameterization, we observed that these distributions can be unstable to train. To cope with this, we utilize a 10x lower learning rate for the parameters of the probability distributions (See Appx. E.3 for details). Nevertheless, finding good ways of learning discrete distributions is an active field of research [21, 22], and advances in this field could be used to further improve the learning of partial equivariances on discrete groups.

**Scaling partial equivariance to large groups.** Arguably the main limitation of G-CNNs with regular representations is their computational and memory complexity, which prevents the use of very large groups, e.g., simultaneous rotation, scaling, mirroring and translations. Partial equivariance is particularly promising for large groups as the network is initialized with a prior towards being equivariant to the entire group, but is able to focus on those relevant to the task at hand. We consider learning partial equivariances on large groups an interesting direction for further research which orthogonal to other advances to scale group convolutions to large groups, e.g., via separable group convolutional kernels [32, 27].

## Acknowledgments and Disclosure of Funding

David W. Romero and Suhas Lohit were supported by Mitsubishi Electric Research Laboratories. David W. Romero is also financed as part of the Efficient Deep Learning (EDL) programme (grant number P16-25), partly funded by the Dutch Research Council (NWO). This work was partially carried out on the Dutch national infrastructure with the support of SURF Cooperative.

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
