# Supplementary Material
## Learning Partial Equivariances from Data

## A    Groups, subgroups, group actions and other group theoretical concepts

**Groups.** Group theory is the mathematical language that describes symmetries. The core mathematical object is that of a *group*, and defines what it means for something to exhibit symmetries. Specifically, a group is a tuple $(\mathcal{G}, \cdot)$ consisting of a set of transformations $\mathcal{G}$, and a binary operation $\cdot$, that exhibit the following properties: (i) closure, i.e., $g_1 \times g_2 = g_3 \in \mathcal{G}, \forall g_1, g_2 \in \mathcal{G}$ (ii) associativity, i.e., $g_1 \cdot (g_2 \cdot g_3) = (g_1 \cdot g_2) \cdot g_3$ for all $g_1, g_2, g_3 \in \mathcal{G}$, (iii) the existence of an identity element $e \in \mathcal{G}$, such that $g \cdot e = e \cdot g = g$, and (iv) the existence of an inverse $g^{-1} \in \mathcal{G}$ for all $g \in \mathcal{G}$.

**Subgroups.** Given a group $(\mathcal{G}, \cdot)$, we say that a subset $\mathcal{H}$ of the group $\mathcal{G}$, is a *subgroup* of $\mathcal{G}$ if this subset also complies to the group axioms under the binary operation $\cdot$. For instance, the set of rotations by $90°$, $\mathcal{H} = \{0°, 90°, 180°, 270°\}$, is a subgroup of the rotation group $\mathrm{SO}(2)$, because it also complies to the closure, associativity, identity and inverse group axioms.

**Group action.** One can define the *action of the group* $\mathcal{G}$ on a set $\mathcal{X}$. This action describes how group elements $g \in \mathcal{G}$ modify the set $\mathcal{X}$ when the transformation is applied. For instance, the action of elements in the group of planar rotations $\theta \in \mathrm{SO}(2)$ on an image $x \in \mathcal{X}$ –written $\theta x$–, depicts how the image $x$ changes when the rotation $\theta$ is applied.

**Lie groups.** A group whose elements form a smooth manifold is referred to as a *Lie group*. Since $\mathcal{G}$ is not necessarily a vector space, we cannot add or subtract group elements –the only operation defined on the group is the binary operation $\cdot$ –. However, if the group is a Lie group, one can link the group $\mathcal{G}$ to a vector space –tangent space at the identity $T_e(\mathcal{G})$–, called the *Lie algebra*. Consequently, one can readily expand group elements on the Lie algebra using a basis $A = \sum_k a^k e_k$ and use these components for calculations. As neural networks work on vector spaces –by means of sums and products–, it is desirable to define convolutional kernels on the Lie algebra as $\psi = \texttt{MLP}: \mathfrak{g} \to \mathbb{R}^{\mathrm{N_{in}} \times \mathrm{N_{out}}}$, where $\mathrm{N_{in}}$ and $\mathrm{N_{out}}$ depict the input and output channels of a convolutional kernel, respectively [16].

**Relevant groups for computer vision applications.** In this work, we consider computer vision applications and thus, are mainly interested in groups that have direct effect on these applications. These groups compose the translation group $\mathrm{T}(2)$, the rotation group $\mathrm{SO}(2)$, the group of rotations and reflections $\mathrm{O}(2)$ and combinations thereof.[5] The actions of these groups can intuitively be understood as the translation, the rotation, and the rotation and reflection of 2D functions, respectively.

These groups can be combined by means of the *semi-direct product* ($\rtimes$) to construct groups that represent combined symmetries. For instance, the 2D roto-translation group $\mathrm{SE}(2) = \mathrm{T}(2) \rtimes \mathrm{SO}(2)$ encompasses symmetries described by both translations and rotations on 2D. Similarly, we can construct a group that describes 2D symmetries given by rotations, translations and reflections $\mathrm{E}(2) = \mathrm{T}(2) \rtimes \mathrm{O}(2)$.[6] Considering equivariance to these groups allows us to construct neural networks that respect the combined symmetries described by them.

## B    Formal treatment of equivariance in partial group convolutions

### B.1    Partial group convolutions from the group $\mathcal{G}$ to a subset $\mathcal{S}$

The partial group convolution from signals on $\mathcal{G}$ to signals on a subset $\mathcal{S}$ can be interpreted as a group convolution for which the output signal outside of $\mathcal{S}$ is set to zero. Consequently, we can calculate the equivariance difference $\Delta_{\mathrm{equiv}}$ in the feature representation, by calculating the difference on the subset $\mathcal{S}$ of a group convolution with a group-transformed input $(\mathcal{L}_w f * \psi)$ and a group convolution with a canonical input proceeded by the same transformation on $\mathcal{S}$, i.e., $\mathcal{L}_w (f * \psi)$.

The equivariance difference $\Delta_{\mathrm{equiv}}^{\mathrm{out}}$ resulting from the effect of considering a subset $\mathcal{S}$ in the output domain of the operation is given by:

---

[5]The names $\mathrm{SO}(2)$, $\mathrm{O}(2)$ are derived from their formal names: Special Orthogonal and Orthogonal group.
[6]The names $\mathrm{SE}(2)$, $\mathrm{E}(2)$ are derived from their formal names: Special Euclidean and Euclidean group.

$$\Delta_{\text{equiv}}^{\text{out}} = \left\| \int_{\mathcal{S}} \mathcal{L}_w(\psi * f)(u)\, \mathrm{d}\mu_{\mathcal{G}}(u) - \int_{\mathcal{S}} (\psi * \mathcal{L}_w f)(u)\, \mathrm{d}\mu_{\mathcal{G}}(u) \right\|_2^2$$

$$= \left\| \int_{w^{-1}\mathcal{S}} (\psi * f)(w^{-1}u)\, \mathrm{d}\mu_{\mathcal{G}}(u) - \int_{\mathcal{S}} (\psi * f)(w^{-1}u)\, \mathrm{d}\mu_{\mathcal{G}}(u) \right\|_2^2$$

$$= \left\| \int_{\mathcal{S}} (\psi * f)(u)\, \mathrm{d}\mu_{\mathcal{G}}(u) - \int_{w\mathcal{S}} (\psi * f)(u)\, \mathrm{d}\mu_{\mathcal{G}}(u) \right\|_2^2$$

$$= \left\| \int_{s_{\min}}^{s_{\max}} (\psi * f)(u)\, \mathrm{d}\mu_{\mathcal{G}}(u) - \int_{ws_{\min}}^{ws_{\max}} (\psi * f)(u)\, \mathrm{d}\mu_{\mathcal{G}}(u) \right\|_2^2$$

$$= \left\| \left( \int_{ws_{\max}}^{s_{\max}} (\psi * f)(u)\, \mathrm{d}\mu_{\mathcal{G}}(u) + \int_{s_{\min}}^{ws_{\max}} (\psi * f)(u)\, \mathrm{d}\mu_{\mathcal{G}}(u) \right) - \right.$$
$$\left. \left( \int_{s_{\min}}^{ws_{\max}} (\psi * f)(u)\, \mathrm{d}\mu_{\mathcal{G}}(u) + \int_{ws_{\min}}^{s_{\min}} (\psi * f)(u)\, \mathrm{d}\mu_{\mathcal{G}}(u) \right) \right\|_2^2$$

$$= \left\| \int_{ws_{\max}}^{s_{\max}} (\psi * f)(u)\, \mathrm{d}\mu_{\mathcal{G}}(u) - \int_{ws_{\min}}^{s_{\min}} (\psi * f)(u)\, \mathrm{d}\mu_{\mathcal{G}}(u) \right\|_2^2$$

From the first line to the second we take advantage of the equivariance property of the group convolution: $(\mathcal{L}_w f * \psi)(u) = \mathcal{L}_w (f * \psi)(u)$, and account for the fact that only the region within $\mathcal{S}$ is visible at the output. We use the change of variables $u = w^{-1}u$ from the second to third line, and specify the boundaries of $\mathcal{S}$, $(s_{\max}, s_{\min})$ from the third to the fourth line. In the fifth line we separate the integration over $\mathcal{S}$ as a sum of two integrals which depict the same range. In the last line, we cancel out the overlapping parts of the two integrals to come to the final result.

In conclusion, the equivariance difference induced by a subset $\mathcal{S}^{(2)}$ on the domain of the output $\Delta_{\text{equiv}}^{\text{out}}$ is given by the difference between the part of the representation that leaves the subset $\mathcal{S}$, and the part that comes to replace it instead. This behaviour is illustrated in Figure 1.

## B.2 Partial group convolutions from a subset $\mathcal{S}^{(1)}$ to a subset $\mathcal{S}^{(2)}$

To isolate the effect of having a group subset as domain of the input signal $f$, we first consider the domain of the output to be the group, i.e., $\mathcal{S}^{(2)} = \mathcal{G}$. The equivariance difference in this case is given by the difference across the entire output representation of the group convolution calculated on an input subset $\mathcal{S}^{(1)}$ with a canonical input $f$, and with a group transformed input $\mathcal{L}_w f$.

The equivariance difference $\Delta_{\text{equiv}}^{\text{in}}$ resulting from the effect of considering a subset $\mathcal{S}^{(1)}$ in the input domain of the operation is given by:

$$\Delta_{\text{equiv}}^{\text{in}} = \left\| \int_{\mathcal{G}} \int_{\mathcal{S}} \psi(v^{-1}u) f(v)\, \mathrm{d}\mu_{\mathcal{G}}(v) \mathrm{d}\mu_{\mathcal{G}}(u) - \int_{\mathcal{G}} \int_{\mathcal{S}} \psi(v^{-1}u) f(w^{-1}v)\, \mathrm{d}\mu_{\mathcal{G}}(v) \mathrm{d}\mu_{\mathcal{G}}(u) \right\|_2^2$$

$$= \left\| \int_{\mathcal{G}} \left[ \int_{\mathcal{S}} \psi(v^{-1}u) f(v)\, \mathrm{d}\mu_{\mathcal{G}}(v) - \int_{\mathcal{S}} \psi(v^{-1}u) f(w^{-1}v)\, \mathrm{d}\mu_{\mathcal{G}}(v) \right] \mathrm{d}\mu_{\mathcal{G}}(u) \right\|_2^2$$

$$= \left\| \int_{\mathcal{G}} \int_{\mathcal{S}} \psi(v^{-1}u) \left[ f(v) - f(w^{-1}v) \right] \mathrm{d}\mu_{\mathcal{G}}(v) \mathrm{d}\mu_{\mathcal{G}}(u) \right\|_2^2$$

In other words, the equivariance difference induced by a subset $\mathcal{S}^{(1)}$ on the domain of the input $\Delta_{\text{equiv}}^{\text{in}}$ is given by the difference in $\mathcal{S}^{(1)}$ between the input $f$, and the part that comes to replace it when the input is modified by a group transformation $w$. This behavior is illustrated in Figure 2.

## C Equivariance property of Monte-Carlo approximations

Consider the Monte-Carlo approximation shown in the main paper:
$$(\psi \,\hat{*}\, f)(u_i) = \sum_j \psi(v_j^{-1} u_i) f(v_j) \bar{\mu}_{\mathcal{G}}(v_j).$$

For a transformed version of the $\mathcal{L}_w f$, we can show that the Monte-Carlo approximation of the group convolution is equivariant in expectation. The proof follows the same steps than Finzi et al. [16]

except that the last step of the proof follows a different reason resulting from the fact that input and output elements can be sampled from different probability distributions.

For a transformed version of the $\mathcal{L}_w f$, we have that:

$$
\begin{aligned}
(\psi \mathbin{\hat{\star}} \mathcal{L}_w f)(u_i) &= \sum_j \psi(v_j^{-1} u_i) f(w^{-1} v_j) \bar{\mu}_{\mathcal{G}}(v_j) \\
&= \sum_j \psi(\tilde{v}_j^{-1} w^{-1} u_i) f(\tilde{v}_j) \bar{\mu}_{\mathcal{G}}(\tilde{v}_j) \\
&\stackrel{d}{=} (\psi \mathbin{\hat{\star}} f)(w^{-1} u_i) = \mathcal{L}_w (\psi \mathbin{\hat{\star}} f)(u_i)
\end{aligned}
$$

From the first to the second line, we use the change of variables $\tilde{v}_j = w v_j$ and the fact that, group elements in the input domain are sampled from the Haar measure for which it holds that $\bar{\mu}_{\mathcal{G}}(v_j) = \bar{\mu}_{\mathcal{G}}(\tilde{v}_j)$. However, from the second to the third line, *we must also assume that this holds for the output domain.* That is, that the probability of drawing $w^{-1} u_j$ is equal to that of drawing $u_j$. We emphasize that this is of particular importance in the partial equivariance setting as this might not be the case in general.

## D    Algorithm for Monte-Carlo approximation of the partial group convolution

---
**Algorithm 1** The Partial Group Convolution Layer
---
1: **Inputs:** position, function-value tuples on the group or a subset thereof $\{v_j, f(v_j)\}$.
2: **Outputs:** convolved position, function-value tuples on the output group subset $\{u_i, (f \mathbin{\hat{\star}} \psi)(u_i)\}$.
3: $\{u_i\} \sim \mathrm{p}(u)$                                           ▷ `Sample elements from p(u)`
4: **for** $u_i \in \{u_i\}$ **do**
5:     $h(u_i) = \sum_j \psi(v_j^{-1} u_i) f(v_j) \bar{\mu}_{\mathcal{G}}(v_j)$            ▷ `Compute group convolution (Eq.` 4`)`
6: **end for**
7: **Return:** $\{u_i, h(u_i)\}$

---

## E    Experimental details

### E.1    Dataset description

**Dataset availability and licensing.** We note that all the datasets used in this paper are publicly available. MNIST is available online under Creative Commons Attribution-Share Alike 3.0 license. CIFAR-10 and CIFAR-100 are available online under MIT license. PatchCamelyon is available online under MIT license.

**Rotated MNIST.** The rotated MNIST dataset [30] contains 62,000 gray-scale 28x28 handwritten digits extracted from the MNIST dataset [31] uniformly rotated on the circle. The dataset is split into training, validation and test sets of 10,000, 2,000, and 50,000 images, respectively.

**CIFAR-10 and CIFAR-100.** The CIFAR-10 dataset [28] consists of 60,000 real-world 32x32 RGB images uniformly drawn from 10 classes divided into training and test sets of 50,000 and 10,000 samples respectively. The CIFAR100 dataset [28] is similar to the CIFAR0 dataset, with the difference that images are uniformly drawn from 100 different classes. For validation purposes, we divide the training dataset of the CIFAR-10 and CIFAR-100 datasets into training and validation sets of 45,000 and 5,000 samples, respectively.

**PatchCamelyon.** The PatchCamelyon dataset [45] consists of 327,000 RGB image patches of tumorous and non-tumorous braset tissues extracted from the Camelyon16 dataset [2], where each patch was labelled as tumorous if the central region of 32x32 pixels contained at least one tomorous pixel as givel by the original annotation in Bejnordi et al. [2]. The dataset is divided into train, validation and test sets of 262,144, 32,768 and 32,768 images, respectively.

### E.2    General remarks

**Hardware.** Our code is written in `PyTorch`. Our experiments were performed on NVIDIA TITAN RTX and V100 GPUs, depending on their availability and the size of the datasets.

**Network specifications.** For almost all the experiments in this paper –except those using the 13-layer CNN of Laine and Aila [29]–, we use the architecture shown in Fig. 3 with an initial lifting convolutional layer followed by 2 ResBlocks with full, partial or regular convolutional layers for Regular G-CNNs, Partial G-CNNs and conventional ($\mathrm{T}(2)$) CNNs. All datasets use a network with 32 feature maps in the hidden layers, Batch Normalization and ReLU.

Table 6: Image recognition accuracy on PatchCam dataset.

| BASE GROUP | NO. ELEMENTS | PARTIAL EQUIV. | CLASSIFICATION ACCURACY ON PATCHCAM (%) |
|---|---|---|---|
| T(2) | 1 | - | 67.59 |
| SE(2) | 8 | ✗ | **89.87** |
|  |  | ✓ | 89.07 |
|  | 16 | ✗ | 89.71 |
|  |  | ✓ | **90.31** |
| E(2) | 16 | ✗ | **89.77** |
|  |  | ✓ | 88.13 |

For MNIST6-M and MNIST6-180, max-pooling is performed after each of the Residual Blocks. In the case of rotMNIST, max-pooling is performed after the lifting convolutional layer and the first group convolutional layer. For CIFAR-10 and CIFAR-100, we use max-pooling after each of the residual blocks. Finally, for PatchCamelyon, we apply max-pooling after the lifting convolution as well as both residual blocks. At the end of the network, a global max-pooling layer is used to create invariant features used for classification. These networks have approximately 460K parameters.

**The continuous group convolutional kernels.** The convolutional kernels of Partial G-CNNs are parameterized as 3-layer SIRENs with 32 hidden units. For the experiments in the main text, we use $\omega_0$=10.0. We compare these to other conventional nonlinearities in Appx. F (Tab. 7). In the case of (partial) group equivariant 13-layer CNNs, the convolutional kernels are constructed as a 3-layer SIREN with 8 hidden units.

### E.3 Hyperparameters and training details

To facilitate replicating our experiments, we provide the list of commands used for our experiments in github.com/merlresearch/partial-gcnn/EXPERIMENTS.md

**Optimization and learning rate schedulers.** Networks on MNIST6-180, MNIST6-M, rotMNIST, CIFAR-10 and CIFAR-100 are trained for 300 epochs and networks on PatchCamelyon are trained for 30 epochs. Furthermore, we utilize a cosine annealing scheduler and combine it with a linear learning rate warm-up for 5 epochs.

**Learning schedulers for the probability distributions** $p(u)$**.** In order to improve the stability of learning the probability distributions on the groups, we utilize a learning rate scheduler similar to that of the main network, i.e., learning rate warm-up for 5 epochs followed by a cosine annealing scheduler, but with a lower base learning rate. Specifically, we use a base learning rate for all probability distributions $p(u)$ of $1e{-}4$.

**Hyperparameters.** We note that all hyperparameters were chosen based on the best performance of the fully equivariant G-CNNs on the validation datasets. The found hyperparameters are subsequently used for the training of our Partial G-CNNs.

We use a batch size of 64 for all networks. In the case of CIFAR-10, CIFAR-100 and PatchCamelyon datasets, we also use a weight decay of $1e{-}4$.

**13-layer CNNs.** Additionally, in the case of 13-layer CNNs we use a dropout rate of 0.3 and train for 200 epochs with batches of size 128. These settings are used on rotMNIST, CIFAR10 and CIFAR100.

## F  Additional Experiments

**Classification results on PatchCamelyon.** Table 6 shows the results obtained for G-CNNs and Partial G-CNNs on the PatchCamelyon dataset [45]. Partial G-CNNs match the performance of G-CNNs in this full equivariant setting. Similar to the rotMNIST case (Fig. 5), the learned probability distributions over the group elements for PatchCamelyon are consistent with Regular G-CNNs.

**Convolution kernels as implicit neural representations.** Next, we validate that SIRENs are better suited to parameterize group convolutional kernels than other alternatives. Tab. 7 shows that SE(2)-CNNs with SIREN kernels consistently outperform SE(2)-CNNs with other parameterizations by a large margin on all the image benchmarks considered. SIREN kernels consistently lead to better accuracy than other existing kernel parameterizations.

Table 7: Comparison of kernel parameterizations.

| MODEL | NO. ELEMENTS | KERNEL TYPE | CLASSIFICATION ACCURACY (%) | | |
|---|---|---|---|---|---|
| | | | ROTMNIST | CIFAR-10 | CIFAR-100 |
| SE(2)-CNN | 4 | ReLU | 96.49 | 59.95 | 28.01 |
| | | LeakyReLU | 94.47 | 56.19 | 27.36 |
| | | Swish | 94.41 | 66.12 | 34.20 |
| | | SIREN | **99.10** | **83.73** | **52.35** |
| | 8 | ReLU | 97.73 | 68.29 | 37.81 |
| | | LeakyReLU | 97.65 | 68.94 | 36.30 |
| | | Swish | 97.72 | 69.20 | 34.10 |
| | | SIREN | **99.17** | **86.08** | 55.55 |
| | 16 | ReLU | 98.49 | 66.84 | 37.72 |
| | | LeakyReLU | 98.53 | 68.01 | 38.29 |
| | | Swish | 98.55 | 65.99 | 37.72 |
| | | SIREN | 99.24 | **86.68** | **51.51** |

Table 8: Results of using additional penalty term to encourage monotonicity in the subset sizes

| GROUP | NO. ELEMENTS | ROTMNIST | CIFAR10 | CIFAR100 |
|---|---|---|---|---|
| SE(2) | 16 | 99.15 | 87.02 | 57.11 |
| E(2) | 16 | 98.41 | 89.00 | 58.85 |

**Enforcing monotonic decreasing group subsets over depth.** Once a Partial G-CNN becomes partial equivariant at some depth, the network is, in general, unable to become fully equivariant at subsequent layers.[7] As a consequence, using fully equivariant layers after a partially equivariant layer does not restore full equivariance.

Based on this observation, one could argue that it is beneficial to impose a monotonically decreasing size to the learned group subsets in order to prevent the at first sight meaningless situation in which the network goes back to larger group subsets. This can be encouraged with an additional *monotonic equivariance loss* term in the training loss, which penalizes bigger subsets at subsequent layers:

$$L_{\text{mon. equiv}} = \sum_{l=1}^{L-1} \left( \gamma_l - \max(\gamma_{l+1}, \gamma_l) \right). \tag{8}$$

Here, $\gamma_l$ represents the limit of the subset learned at the $l$-th layer.

Interestingly, we find that due to the reasons explained in Sec. 6 imposing a monotonic decrease on the learned subsets leads to slightly worse performance than an unconstrained model (see Tabs. 8, 3).

# G  Broader social impact

This work is fundamental and mathematical in nature. We believe it does not pose any immediate harm to society. However, the exact applications of these ideas could have negative impact and thus, care should be taken when using these ideas in machine learning. One motivation of this paper is to make deep networks more robust to nuisance factors and can hopefully be safer than earlier works.

---

[7]An exception to this rule is when the a layer goes back to the original input space, i.e., $\mathcal{S}^{(2)} = \mathcal{X}$, and the immediately subsequent layer goes back to the full group. This case is equivalent to performing a projection along a group axis, and going back to the full group afterwards, i.e., a lifting convolution.