# OpenReview forum: "Learning Partial Equivariances From Data"
_NeurIPS.cc/2022/Conference — NeurIPS 2022 Accept_

### Official Review · Reviewer_BwFs · 2022-06-29

**Rating:** 6
**Confidence:** 3
**Soundness:** 2 fair
**Presentation:** 2 fair
**Contribution:** 3 good

**Summary:**

This paper analyzes "the extent of equivariance"  when the input and output domains are the whole groups or the subsets of the group. Based on the analysis, the authors propose the partial G-CNN by learning the probability distribution $p(u)$ over the group in every neural network layer. Their method works for the discrete group, continuous group, and combinations. The experiments show that partial G-CNN can learn the different sets of transformations depending on the task.

**Questions:**

1.Line 237 replaces the [15] with a convolution, but I'm confused about what $v^{-1}u$ is, which is the same question in weakness 2.

2.In the experiments, there are some tasks we already know; for example, MNIST6-180 and MNIST6-M, can we derive the fixed out domain but not learn the out domain layerwise to get a better result than conducting convolution over the whole group. I'm a little confused about how to leverage the analysis on partial equivariance quantitively.


**Limitations:**

The authors list the reasonable limitation in the paper and the broader social impact in the supplementary materials.

**Strengths And Weaknesses:**

Strength:
1.The problem, transformations in the data might be better represented by a subset of the group rather than the whole group, proposed in the paper is novel and is not discussed in previous work in equivariance. This problem deserves more exploration and attention.

2.The analysis of group convolution for different input and output domains is sound, and the method to obtain the output domain by learning the probability distribution over the group is novel.

 3.The paper is well organized, and the images in the paper help audience understand the paper easily.

Weakness:
1.The paper proposes a method of learning the probability distribution of discrete groups, continuous groups, and the combination. Could the authors show more experiments on the group other than SE(2), E(2), or the flipping group?

2.I'm a little confused about the equation (3); what is $v^{-1}u$ since v is in the domain that $G$ acting on and $u$ is in G? How to get $v^{-1}$ and how does it act on the group element? Correct me if I'm wrong.

3.There is a strong baseline that I think the authors should compare with; that is, the equivariant network over the whole group by replacing the max-pooling layer with the flatting operation and the MLP layer.  The conventional equivariant networks are equivariant to the whole group, and if we don't do pooling over the group dimension, then the transformation information is kept; for example, here, the permutation of features for "6" and "9" are different, which also can help the network to classify 6 and 9.

4.There are several typos in the paper, especially the reference to the figure and the table. I suggest the authors could check the misalignment.

---

> ### Author Response · Authors · 2022-07-29
> **First Response -- reviewer BwFs**
>
> Dear reviewer BwFs,
>
> First of all, we would like to thank you very much for your review. We sincerely appreciate the time you spent in evaluating our work, and very much appreciate your comments.
>
> Here we will answer to your questions, comments and concerns:
>
> **[Weaknesses]**
>
> **1.The paper proposes a method of learning the probability distribution of discrete groups, continuous groups, and the combination. Could the authors show more experiments on the group other than SE(2), E(2), or the flipping group?** Which other groups would you like to see explored? We agree that learning partial equivariances on larger groups and larger datasets is an important and interesting research direction. Unfortunately, we have constrained computational resources, which prevents us from training G-CNNs on large groups or large datasets.
>
> Nevertheless, to highlight the importance of this endeavor, we have stated this explicitly in the current future work section (Sec. 7) of our work and provide insights on how the scalability issue can be tackled.
>
> **2.I'm a little confused about the equation (3); what is $v^{-1}u$  since $v$ is in the domain that $G$ acting on and $u$ is in G? How to get $v^{-1}$ and how does it act on the group element? Correct me if I'm wrong.**
> In short, this is equivalent to the more familiar case, e.g., in Cohen & Welling, (2016), where it is written $u^{-1}v$. The difference is how the action on $X$ is defined. In that paper, it is assumed that the group acts on $X$ from the left, whereas in our derivations it is assumed that it acts on it from the right. If you agree that clarity would be improved if we stick to left actions, we can change this throughout the document.
>
> **3.There is a strong baseline that I think the authors should compare with; that is, the equivariant network over the whole group by replacing the max-pooling layer with the flatting operation and the MLP layer. The conventional equivariant networks are equivariant to the whole group, and if we don't do pooling over the group dimension, then the transformation information is kept; for example, here, the permutation of features for "6" and "9" are different, which also can help the network to classify 6 and 9.** This is true. We actually compare to your proposed baseline as well as to a G-CNN on which only the last layer is a T(2) layer (thus breaking equivariance) (see Tab. 4). Although these methods are also able to solve the toy tasks, they underperform on rotMNIST, CIFAR10 and CIFAR100.
>
> We have now rewritten the experimental section to highlight the differences and benefits of Partial G-CNNs.
>
> **4.There are several typos in the paper, especially the reference to the figure and the table. I suggest the authors could check the misalignment.** This has been fixed now.
>
> **[Questions]**
>
> **1.Line 237 replaces the [15] with a convolution, but I'm confused about what  is, which is the same question in weakness 2.** Please see the previous answers.
>
> **2.In the experiments, there are some tasks we already know; for example, MNIST6-180 and MNIST6-M, can we derive the fixed out domain but not learn the out domain layerwise to get a better result than conducting convolution over the whole group. I'm a little confused about how to leverage the analysis on partial equivariance quantitively.** This is an interesting question. In principle, for multilayered nets, it isn't easy to derive the optimal domains as equivariance can be restricted in several dfferent ways throughout the network to arrive to the same results. With that being said, we do know the optimal for fully equivariant / invariant tasks: staying fully equivariant throughout the network. We observe that this is learned by the model on rotated MNIST (Fig. 5).
>
> For the partial equivariant case, what we could do is take a single layer and see whether the learned subset matches the expected partial equivariance (e.g., rotations in $[-90^\circ, 90^\circ]$ for MNIST6-180. If the reviewer agrees, we can include this additional study in our paper.
>
> With that being said, we would like to emphasize that, as shown in Fig 4, Partial G-CNNs do predict the (almost) optimal response in partial equivariant settings as well.
>
> **[Final words]** We hope that these responses clarify your questions and concerns. Please let us know if you have any follow-up / additional questions.
>
> Best regards,
>
> The Authors

---

> > ### Comment · Reviewer_BwFs · 2022-08-08
> > **Line 237 issue**
> >
> > If it acts on $X$ from the right, then $L_\omega(\mu)$ is not $L(\omega^{-1}\mu)$, correct me if I'm wrong.

---

> > > ### Author Response · Authors · 2022-08-09
> > > **Response Line 237 Issue**
> > >
> > > That is right. What you describe is the left action of the group.
> > >
> > > If we define the right action as $\mathcal{L}_{u}(v) = (vu^{-1})$ we can also express this as a left action with some algebra. As explained -among others- in Wikipedia,
> > >
> > > > Because of the formula $(gh)^{−1}=h^{-1}g^{-1}$, one can construct a left action from a right action by composing with the inverse operation of the group.
> > >
> > > In other words, we can also describe the right action $\mathcal{L}_{u}(v)= (vu^{-1})$ as $(v^{-1}u)$ like in the paper.
> > >
> > > We understand this can be confusing. We used this notation to follow Finzi et al. 2020 closely. However, if you agree that following the (more conventional) left action is clearer, we will happily change this throughout the paper. We note that the selection between left and right actions  does not change the core contribution of the paper, and simply involves swapping the order of these operations.
> > >
> > > Best regards,
> > >
> > > The authors.

---

> ### Comment · Reviewer_BwFs · 2022-08-08
> **Post-rebuttal comments**
>
> The authors clarified most of my questions and concerns, and I would maintain my rating mainly because of the experiment issue, which the other reviewers also mentioned.

---

> > ### Author Response · Authors · 2022-08-09
> > **Response to post-rebuttal comments**
> >
> > Dear reviewer,
> >
> > thank you very much for your response.
> >
> > We are currently running experiments with the 13-layer CNN architecture from Laine and Aila, 2016 used in Augerino. We are uncertain we will have these results during the rebuttal period, but we will make sure these results appear in the paper for CIFAR10, CIFAR100 and STL10.
> >
> > Best regards,
> >
> > The authors.

---

### Official Review · Reviewer_9rGm · 2022-07-11

**Rating:** 6
**Confidence:** 4
**Soundness:** 3 good
**Presentation:** 2 fair
**Contribution:** 3 good

**Summary:**

The paper presents a strategy to learn partial equivariances from the data, in the setting of image classification. This approach allows to learn different degrees of equivariance, from the full E(2) to identity map. The idea is to parametrise the group elements via a probability distribution, which is learned via montecarlo sampling of convolution at different activation layers of a deep network. Results on toy and real data show good results, competitive with standard translation equivariant architectures.

**Questions:**

- It is unclear to me what is the base model structure used in T(2) settings. Authors mention the use of ResNets, but what are base models? De facto standard architectures are achieving lots better on CIFAR, but the current setting is a bit obscure to me. Not that I want to see numbers better than SOTA, of course, but I cannot understand the comparison. Maybe some details, a number of parameters to compare, could help.
- In papers [1] numbers in Tab 4 for the CIFAR are competitive and the model is also simple. Models [1] and [2], which are similar, can be both extended to known partial equivariances (but not learned, which is the nugget here). Maybe those references could serve some comparisons or discussions, if time and space allow.
- L220-229. Could it be that some group elements are not independent? e.g. vertical flips would prevent some specific rotations? Is there a way to cope with it?
- Sec  5: Is not clear what "mirror CNN" are.
- The number of elements vary always as 8 or 16. I am fine with that, but there does not seem to be a good explanation / motivation / justification.
- There are some missing references (??) in the latex in the main text (L331) and supp mat (L53).





[1] Zhou et al, oriented response network, CVPR 17
[2] Marcos et al, rotation equivariant vector field networks, ICCV 2017

**Limitations:**

Limitations are addressed in Sec 7, which concludes the paper. I think those are enough and relevant. Some points raised above could be also discussed there, as probably not all my comments can be addressed.

**Strengths And Weaknesses:**

### Strengths

- The paper makes an interesting point, and indeed I think that allowing models to learn partial equivariances to some sub-groups would improve or worst case match performance of standard models. At least, this seem what results are pointing at, even if datasets are maybe not so convincing.
- The proposed solution, although maybe a bit hidden in the cumbersome formulations and description, is simple yet effective, and seem to be a very interesting solution. I find interesting the fact that the model can learn different levels of equivariance to predefined groups, at different layers. I would be interesting in future work potentially, to constrain specific layers to not have this level of flexibility, and to see if others compensate of if the learning breaks down. In Fig. 5 it seem that the model naturally learns to not be equivariant in layers 3/4 and it would be interesting to see why and what are the limits.

### Weaknesses
- The theoretical developments in the paper are complex, and as such, they should be followed by clear and exemplifying text, which is not always the case. I found these parts hard to follow, for messages that essentially can be simplified to high-level descriptions and reader sent to references.
- This would leave some room to describe better the Monte Carlo estimation (sec. 4) and to move some details and results from the supplementary material to main text. I feel for some of that, the supp mat is not the right place (eg App A, with a clear description of what SE(2), T(2), E(2) are, App. D, some of E and F).
- I feel CIFAR datasets are generally not so easy to use for further generalisation on realistic cases. MNIST even less. I unfortunately deeply miss some more realistic image classification dataset. ALthough is interesting to see results are improving by learning partial equivariances (some tilts, balancing horizontal flips in training set, pose for some classes, etc), I feel that those results alone are not enough to have general claims.
- Figure 1 is impossible to read (resolution, colour), captions in general are not great.
- Section 3.3 is unclear overall, maybe needs some reworking
- In the experiments, I miss an illustration of a loss curve, or something that shows how the training converges. It is also underlined in Sec 7, that the training is unstable and tough, so I wonder, how the authors coped with it. If any trick, weird scheduling, or anything, were used, I think it would be fair to mention it.

---

> ### Author Response · Authors · 2022-07-29
> **First Response -- reviewer 9rGm**
>
> Dear reviewer 9rGm,
>
> First of all, we would like to thank you very much for your review. We sincerely appreciate the time you spent in evaluating our work, and very much appreciate your comments.
>
> Here we will answer to your questions, comments and concerns:
>
> **[Strengths]**
>
> **although maybe a bit hidden in the cumbersome formulations and description, is simple yet effective, and seem to be a very interesting solution.** We have rewritten parts of our work to try to make our paper more understandable. We hope this has helped. In addition, we have included all terms that are used later on in the paper in Appx. A, and linked it to the beginning of Sec. 2.
>
> **I would be interesting in future work potentially, to constrain specific layers to not have this level of flexibility, and to see if others compensate of if the learning breaks down.** Do you mean constraining, for instance, the first layers of the network to be fully equivariant? We haven't tried this explicitly but we would expect the network to break equivariance in the subsequent layers. Note that the first layer --the lifting-- is actually fully equivariant and despite this, following layers break equivariance if advantageous.
>
> **In Fig. 5 it seem that the model naturally learns to not be equivariant in layers 3/4 and it would be interesting to see why and what are the limits.** Indeed! This is very interesting. We have some experiments in which we encourage the subsets to be monotonically smaller over depth (Appx. F). However, this does not lead to better results.
>
> **[Weaknesses]**
>
> **The theoretical developments in the paper are complex, and as such, they should be followed by clear and exemplifying text, which is not always the case. I found these parts hard to follow, for messages that essentially can be simplified to high-level descriptions and reader sent to references.** In a previous version of our work we did exactly this. Nevertheless, reviewers found it not to be formal and specific enough. Hence, changes needed to be made. What we can do is to provide a (less exact but)  intuitive high level explanation of our core ideas in the Appendix. As an alternative, we can promise to write a blog / make a video on which the core concepts are explained on a high level and link it to the paper.
>
> **This would leave some room to describe better the Monte Carlo estimation (sec. 4) and to move some details and results from the supplementary material to main text. I feel for some of that, the supp mat is not the right place (eg App A, with a clear description of what SE(2), T(2), E(2) are, App. D, some of E and F).**
> We have reorganized the paper a bit and included some of these details in the main text. We have also provided a more complete description of the Monte Carlo approximation in the main text. Please let us know if you find this sufficient.
>
> **I feel CIFAR datasets are generally not so easy to use for further generalisation on realistic cases. MNIST even less. I unfortunately deeply miss some more realistic image classification dataset. ALthough is interesting to see results are improving by learning partial equivariances (some tilts, balancing horizontal flips in training set, pose for some classes, etc), I feel that those results alone are not enough to have general claims.**
> We would like to highlight that we provided results on 4 benchmark datasets (rotMNIST, CIFAR10 , CIFAR100, PathCamelyon)  and 2 toy datasets. With that being said, we entirely understand your point. Nevertheless, we unfortunately have constrained computational resources, which prevents us from training large G-CNNs or training on large datasets.
>
> We agree that learning partial equivariances on larger groups and larger datasets is an important and interesting research direction. To highlight the importance of this endeavor, we have stated this explicitly in the current future work section (Sec. 7) of our work and provide insights on how the scalability issue can be tackled.
>
> **Figure 1 is impossible to read (resolution, colour), captions in general are not great.**
> We have done some modifications please let us know if you find it is sufficient now.
>
> **Section 3.3 is unclear overall, maybe needs some reworking** We have partially rewritten this section. We hope readability has improved.

---

> > ### Author Response · Authors · 2022-07-29
> > **First Response -- reviewer 9rGm -- continuation**
> >
> > **In the experiments, I miss an illustration of a loss curve, or something that shows how the training converges. It is also underlined in Sec 7, that the training is unstable and tough, so I wonder, how the authors coped with it. If any trick, weird scheduling, or anything, were used, I think it would be fair to mention it.** The training curves seem pretty normal in comparison to other CNNs. We can add these to the Appendix if so desired. We believe that the desire to see the loss curves comes from the second point in your comment.
> >
> > We realized that we over-exaggerated this problem in our original submission. We stated this as an important problem not to understate the difficulty of learning discrete distributions --which is an active research field--.
> >
> > To account for this, we have now rewritten this limitation paragraph and provided details on how the distributions are actually learned (See Sec.7 & Appendix). In a nutshell, we make the training of these distributions more stable by using a lower learning rate for the parameters of the distributions.
> >
> > **[Questions]**
> >
> > **It is unclear to me what is the base model structure used in T(2) settings. Authors mention the use of ResNets, but what are base models? De facto standard architectures are achieving lots better on CIFAR, but the current setting is a bit obscure to me. Not that I want to see numbers better than SOTA, of course, but I cannot understand the comparison. Maybe some details, a number of parameters to compare, could help.** We use the same ResNet architecture with 2 residual blocks for all groups. The baseline model is shown in Figure 3 & explained briefly at the beginning of Sec 5. We apologize that the information was not sufficient. In the current version of the paper we have included additional details in Sec. 3.5, Sec. 5 and the appendix (Appx. E, F).
> >
> > **In papers [1] numbers in Tab 4 for the CIFAR are competitive and the model is also simple. Models [1] and [2], which are similar, can be both extended to known partial equivariances (but not learned, which is the nugget here). Maybe those references could serve some comparisons or discussions, if time and space allow.** Thank you very much for the references. We will include them in our discussion. Regarding the models, we could provide additional results with other networks. However, we believe that that would not change the current pitch of the paper. We can explore other neural architectures if the reviewer agrees with the added value of these experiments.
> >
> > **L220-229. Could it be that some group elements are not independent? e.g. vertical flips would prevent some specific rotations? Is there a way to cope with it?**
> > This is a very interesting and relevant question. It depends on how one formulates the probability distributions on each dimension. In our experiments we utilize flips along the y axis when using reflections. This is important as flips around other axes (e.g., the horizontal axis) are then defined as a combination of a rotation and a flip. If we constrain then constrain rotations to certain angles, then not all possible flips can be performed. Given that flips along the vertical axis are more likely to appear in images, we selected this to be matching with the identity rotation.
> >
> > This is an interesting point worth investigating. However, this is a complex topic and, in our eyes, worth its own paper, as it leads to the question of how to formulate different kinds of probability distributions on groups and what properties they end up having.
> >
> > We will include a word on this in the limitations of our work.
> >
> > **Sec 5: Is not clear what "mirror CNN" are.** We have now made this clear in the paper. Mirror-CNN is a reflection equivariant CNN.
> >
> > **The number of elements vary always as 8 or 16. I am fine with that, but there does not seem to be a good explanation / motivation / justification.** Yes there is. The simple one is that previous works normally use 4 (90 degrees), 8 (45 degrees) and 16 (22.5 degrees) elements in their discrete approximations. The more complete response is that the higher the number of elements, the better the estimation of the integral along the group axis will be, and thus, the more exact equivariance will be. For works which approximate the continuous work with group discretizations, e.g., Weiler & Cesa (2019), the number of samples relates to the number of transformations to which the model will be exactly equivariant. A nice illustration of this behavior can be observed in Fig. 5 of Weiler et al. (2018).
> >
> > [Weiler et al. *Learning Steerable Filters for Rotation Equivariant CNNs*, CVPR, 2018]
> >
> > **There are some missing references (??) in the latex in the main text (L331) and supp mat (L53).** These have been solved now.
> >
> > **[Final words]** We hope that these responses clarify your questions and concerns. Please let us know if you have any follow-up / additional questions.
> >
> > Best regards,
> >
> > The Authors

---

> > > ### Comment · Reviewer_9rGm · 2022-08-09
> > > **Post rebuttal**
> > >
> > > Thanks for the follow up and clarifications. I feel like most comments have been addressed, and I raise the score to weak accept.

---

> > > > ### Author Response · Authors · 2022-08-09
> > > > **Post rebuttal response**
> > > >
> > > > Dear reviewer,
> > > >
> > > > We thank you very much for your reply.
> > > >
> > > > Best regards,
> > > >
> > > > The authors

---

### Official Review · Reviewer_5hXP · 2022-07-11

**Rating:** 5
**Confidence:** 4
**Soundness:** 3 good
**Presentation:** 2 fair
**Contribution:** 2 fair

**Summary:**

The paper introduces a way to learn partial equivariances from data by using a Monte Carlo group convolution where the samples are taken from a 1D uniform distribution with its bounds learned via the reparametrization trick. The method is shown to work on some tasks where equivariant models fail, like when the task is to detect a specific transformation. It also seems to compare favorably against somewhat weak baselines on natural image classification tasks (CIFAR10/100).


**Questions:**

- The patterns shown in Fig 5 are intriguing and are the most interesting result of the paper in my opinion. I wouldn't expect the bounds to narrow down in middle layers and widen up at the final. I suggest investigating this further as it might provide useful insights.

- There is a mismatch between the accuracy for SE(2), 16, Partial on CIFAR10 in tables 3 and 4.

- L85- states "The group convolution requires integration over a continuous domain. Hence, it cannot be computed in finite time and must be approximated.". This is not true, there are many integrals over continuous domains that can be computed exactly in finite time, including group convolutions under certain conditions (for example, between bandlimited functions on compact groups).

- L331- missing reference: "Tab. ??".


**Limitations:**

Adequately addressed.

**Strengths And Weaknesses:**

# Strengths

The submission addresses an interesting and hard problem, which is how to learn which symmetries are present in the data and how to exploit them. It shows some intriguing results when allowing each layer of the model to be equivariant to a different subset of the euclidean group.

# Weaknesses

1) The major weakness I see is that the main ideas of the submission are similar to Augerino [1], which also uses the reparametrization trick to learn the bounds of a uniform distribution to which the model should be invariant. The main difference is that Augerino uses the learned distribution to augment the input, whereas the proposed method applies the idea to the samples of the Monte Carlo group convolution from Finzi et al [2].

2) A second difference with respect to Benton et al [1] is that the submission introduces a way to learn discrete distributions; however, this seem only evaluated on the simplest 2-element group of reflections, and even in that case it seems unstable to train as stated in L373. I don't understand the motivation of the approach for the discrete case in L206-, wouldn't it make more sense to one  probability mass for each point and sample several points from the resulting distribution?

3) The comparison in Table 2 doesn't seem fair to Augerino, as they report an accuracy of 93.81% on CIFAR10, versus <85% reported in the submission. I understand that the architectures and training scheme are different, but the results would be much more convincing if they followed the baseline's hyperparameters and showed improvements over those numbers. Moreover, Benton et al [1] also evaluate on more interesting tasks such as SE(3) invariance in molecular property prediction and color-space invariance on STL-10. The submission would be made stronger by including these and/or other tasks that involve groups other than E(2) subgroups.

4) Some results in tables 2, 3, 4 are very close to one another, so it is unclear if the differences are statistically significant. I recommend running each training job several times and including confidence intervals.

# References

[1] Benton et al, "Learning Invariances in Neural Networks", NeurIPS'20.

[2] Finzi et al, "Generalizing Convolutional Neural Networks for Equivariance to Lie Groups on Arbitrary Continuous Data", ICML'20.

---

> ### Author Response · Authors · 2022-07-29
> **First Response -- reviewer 5hXP**
>
> Dear reviewer 5hXP,
>
> First of all, we would like to thank you very much for your review. We sincerely appreciate the time you spent in evaluating our work, and very much appreciate your comments.
>
> Here we will answer to your questions, comments and concerns:
>
> **[Weaknesses]**
>
> **The major weakness I see is that the main ideas of the submission are similar to Augerino [1], which also uses the reparametrization trick to learn the bounds of a uniform distribution to which the model should be invariant. The main difference is that Augerino uses the learned distribution to augment the input, whereas the proposed method applies the idea to the samples of the Monte Carlo group convolution from Finzi et al [2].**
> We do not consider this to be a weakness of our work. It is true that we use insights from these previous works --which we properly cite and acknowledge--. However, as outlined on the paper, our approach has several advantages over Augerino. For once, Augerino does not consider equivariance but only invariance and it is unclear how representations within the network behave. In addition, Augerino does not allow the learning of equivariant representations for invariant tasks such as classification. Contrarily, Partial G-CNNs allow both, and focus on a more difficult problem: that of learning and exploiting equivariance to symmetries directly from data.
>
> Regarding Finzi et al. 2020, we also provide some improvements. We promote the use of better kernel parameterizations, and replace their isotropic lifting by a non-isotropic one a la Cohen & Welling, 2016. The latter is important because isotropic lifting induces the following group conv. layer to learn invariant representations instead of an equivariant one. This phenomenon is briefly explained in Lengyel & van Gemert (2021) for group convolutional layers. We have modified Section 3.5. to make these differences clearer.
>
> [Lengyel & van Gemert, *Exploiting Learned Symmetries in Group Equivariant Convolutions*, ICIP 2021]
>
> **A second difference with respect to Benton et al [1] is that the submission introduces a way to learn discrete distributions; however, this seem only evaluated on the simplest 2-element group of reflections...** We consider the reflection group as it has important applications for computer vision. Apart from the reflection and permutation group, we are not aware of any other relevant discrete group for machine larning applications. It would be interesting to consider the permutation group in graph applications. However, the permutation group is very large and it is --to the best of our knowledge-- not feasible to construct permutation equivariant CNNs with regular representations.
>
> We do have experiments for cyclic and dihedral groups (subgroups of SO(2) and O(2)), which we could report in our paper. We did not add these results to our paper as we consider the continuous SO(2) and O(2) groups, and these encapsulate these discrete groups --which are actually used to approximate their continuous counterparts when G-CNNs are unable to work on continuous spaces--. Nevertheless, we can add these results if the reviewer sees the added value of having them in our submission.
>
> **...and even in that case it seems unstable to train as stated in L373.** We realized that we over-exaggerated this problem in our original submision. Our method is actually able learn to these distributions, but we stated this as an important problem not to understate the difficulty of learning discrete distributions --which is an active research field--.
>
> To account for this, we have now rewritten this limitation paragraph and provided details on how the distributions are actually learned (See Sec.7 & Appendix). In a nutshell, we make the training of these distributions more stable by using a lower learning rate for the parameters of the distributions.
>
> **I don't understand the motivation of the approach for the discrete case in L206-, wouldn't it make more sense to one probability mass for each point and sample several points from the resulting distribution?** That is also a possibility. However, as explained in the paper, this would need us to learn the probability of each particular combination independently. This is not only costly in terms of parameters --$2^n$ instead of $n$-- but also makes tuning the corresponding probabilities more difficult. The later results from having a much lower probability of seeing each of these combinations to update the corresponding parameters --proportional to $\frac{1}{2^n}$ as opposed to $\frac{1}{n}$--.

---

> > ### Author Response · Authors · 2022-07-29
> > **First Response -- reviewer 5hXP -- continuation**
> >
> > **I understand that the architectures and training scheme are different, but the results would be much more convincing if they followed the baseline's hyperparameters and showed improvements over those numbers. Moreover, Benton et al [1] also evaluate on more interesting tasks such as SE(3) invariance in molecular property prediction and color-space invariance on STL-10. The submission would be made stronger by including these and/or other tasks that involve groups other than E(2) subgroups.** We entirely understand and support your point. However, we unfortunately have constrained computational resources, which prevents us from training large G-CNNs, for large groups or on larger datasets .
> >
> > We agree that learning partial equivariances on larger groups is an important and interesting research direction. In particular it provides a prior on the model to respect certain equivariances and then leaves it up to the model to scrap those that are irrelevant or harmful for the task at hand. To highlight the importance of this endeavor, we have stated this explicitly in the current future work section (Sec. 7) of our work and provide insights on how the scalability issue can be tackled.
> >
> > **Some results in tables 2, 3, 4 are very close to one another, so it is unclear if the differences are statistically significant. I recommend running each training job several times and including confidence intervals.** We will include standard deviations over 5 runs to our reported results.
> >
> > **[Questions]**
> > **The patterns shown in Fig 5 are intriguing and are the most interesting result of the paper in my opinion. I wouldn't expect the bounds to narrow down in middle layers and widen up at the final. I suggest investigating this further as it might provide useful insights.** We were thinking about adding a figure illustrating how the hidden representations change in such a setting. However, this will be different as iis ilustrated in the final layer of Partial G-CNNs (Fig. 5). What do you think? Do you think this adding these would add insights to the paper? If you have other ideas in mind, please let us know!
> >
> > **There is a mismatch between the accuracy for SE(2), 16, Partial on CIFAR10 in tables 3 and 4.** This will be fixed.
> >
> > **L85- states "The group convolution requires integration over a continuous domain. Hence, it cannot be computed in finite time and must be approximated.". This is not true, there are many integrals over continuous domains that can be computed exactly in finite time, including group convolutions under certain conditions (for example, between bandlimited functions on compact groups).** You are right. We have modified this in our current submission. We now emphasize that this is only the case in general.
> >
> > **L331- missing reference: "Tab. ??".** This has been fixed.
> >
> > **[Final words]** We hope that these responses clarify your questions and concerns. Please let us know if you have any follow-up / additional questions.
> >
> > Best regards,
> >
> > The Authors

---

> ### Comment · Reviewer_5hXP · 2022-08-03
> **Post-rebuttal comments**
>
> I appreciate the responses and I intend to maintain my recommendation.
>
> Regarding 1), the advantages wrt the baselines would be much more convincing with the fair comparisons I mentioned in 3), and other reviewers also raised experimental weaknesses. I understand that lack of resources might be an issue, but CIFAR10 is quite small, as is the model used in Augerino, so I imagine it would train quickly on a single GPU. Please correct me if I'm wrong.
>
> Regarding Fig 5, yes, analyzing the feature maps might be the way to go. Other reviewer also noted this interesting fact and I believe explaining it or even raising some hypothesis would be welcome.

---

> > ### Author Response · Authors · 2022-08-09
> > **Response to post-rebuttal comments**
> >
> > Dear reviewer,
> >
> > thank you very much for your response.
> >
> > We are currently running experiments with the 13-layer CNN architecture from Laine and Aila, 2016 used in Augerino. We are uncertain we will have these results during the rebuttal period, but we will make sure these results appear in the paper for CIFAR10, CIFAR100 and STL10.
> >
> > With regard to the analysis, thank you for the suggestions. We will include plots showing how the representations on the last group convolutional layer change for different transformations on the input analogously to Figure 5. In the Discussions section (Sec. 6) we outline our main hypothesis, which is that the group dimension can serve to encode mappings that depend on the section of the group that is observed. We will update this as soon as the analysis is ready.
> >
> > Best regards,
> > The authors.

---

### Meta-Review · Area_Chair_Vasg · 2022-08-25

**Recommendation:** Accept
**Confidence:** Certain

**Metareview:**

All three reviewers lean toward acceptance. After the author-reviewer discussion, reviewers find most of their concerns clarified. Concerns about several experimental issues remain, and the authors provided reasonable responses. After careful consideration, AC recommends accepting the paper.

**Award:**

No

---

### Decision · Program_Chairs · 2022-09-14

Accept